# Investigation of Colored Film Indicators for the Assessment of the Occasional Radiation Exposure

**DOI:** 10.3390/gels9030189

**Published:** 2023-02-28

**Authors:** Linas Kudrevicius, Diana Adliene, Judita Puiso, Aurimas Plaga

**Affiliations:** Physics Department, Kaunas University of Technology, 44249 Kaunas, Lithuania

**Keywords:** colored films, X-ray irradiation, radiation exposure indicators

## Abstract

Occupational radiation exposure monitoring is well-established in clinical or industrial environments with various different dosimeter systems. Despite the availability of many dosimetry methods and devices, a challenge with the occasional exposure registration, which may occur due to the spilling of radioactive materials or splitting of these materials in the environment, still exists, because not every individual will have an appropriate dosimeter at the time of the irradiation event. The aim of this work was to develop radiation-sensitive films—color-changing radiation indicators, which can be attached to or integrated in the textile. Polyvinyl alcohol (PVA)-based polymer hydrogels were used as a basis for fabrication of radiation indicator films. Several organic dyes (brilliant carmosine (BC), brilliant scarlet (BS), methylene red (MR), brilliant green (BG), brilliant blue (BB), methylene blue (MB) and xylenol orange (XiO)) were used as a coloring additives. Moreover, PVA films enriched with Ag nanoparticles (PVA-Ag) were investigated. In order to assess the radiation sensitivity of the produced films, experimental samples were irradiated in a linear accelerator with 6 MeV X-ray photons and the radiation sensitivity of irradiated films was evaluated using UV–Vis spectrophotometry method. The most sensitive were PVA-BB films indicating 0.4 Gy^−1^ sensitivity in low-dose (0–1 or 2 Gy) range. The sensitivity at higher doses was modest. These PVA-dye films were sensitive enough to detect doses up to 10 Gy and PVA-MR film indicated stable 33.3% decolorization after irradiation at this dose. It was found that the dose sensitivity of all PVA-Ag gel films varied from 0.068 to 0.11 Gy^−1^ and was dependent on the Ag additives concentration. Exchange of a small amount of water with ethanol or isopropanol caused the enhancement of radiation sensitivity in the films with the lowest AgNO_3_ concentration. Radiation-induced color change of AgPVA films varied between 30 and 40%. Performed research demonstrated the potential of colored hydrogel films in their applications as indicators for the assessment of the occasional radiation exposure.

## 1. Introduction

Recently, some fields and applications are not imaginable without using ionizing radiation. The most important fields are medicine, nuclear power and industrial applications. Medical diagnostic and imaging using X-rays, cancer treatment using linear accelerator, gamma knife, brachytherapy and others, nuclear medicine, including production of radionuclides, diagnostics and treatment of patients with radionuclides, as well as nuclear power generation or industrial radiation processing of materials, radiation sterilization of food, etc. This supposes the need for careful monitoring of all mentioned activities in order not to crate the harm for individuals and environment. There are different methods and various types of detectors that are applied for the detection and measurement of occupationally exposed workers and quality assurance programs are implemented. However, despite of existing exposure monitoring methods and devices used, for the detection and quantification of radiation leakage, occasional radiation exposure when working with them (for example, hands of nuclear medicine department staff) still may create problems due to the fact that not every individual may have a proper detector at the time of an irradiation event.

Different dose registration systems based on radiation-induced physical phenomena and chemical reactions in detector materials are used, such as gas filled detectors (G-M counters), optically-stimulated luminescence (OSL), thermoluminescence (TL) and radioluminescence (RL) detectors [1], or chemical dosimeters [2,3,4] and others. A number of publications discussing hydrogel-based sensors (Fricke, radio-chromic, radio-fluorogenic gel dosimeters) [5] has increased significantly during the last decades due to the fact that gel sensors are active detectors and indicate irradiation-specific visual changes of their optical characteristics.

Dosimeters may have various dynamic range depending on their chemical composition, concentration, readout method, overall calibration and field of application. Previously, color dosimetry systems encountered three main problems: the necessity to apply a high irradiation dose (kGy range) for detectable color change, short chemical stability of the dosimeter after preparation and irradiation and the necessity of calibration for each newly prepared chemical composition. Well-known radiochromic systems that are used for sterilization or radiation processing of materials have a dynamic range up to 5–50 kGy [6,7,8] or 0.15–4.00 kGy [9,10], Trichloroethylene- and cresol-red-based dosimeters are used to identify gamma irradiation in the dose range between 1 and 12 kGy [11] and nitro blue tetrazolium chloride (NBT) films with polyvinyl alcohol or butyral—even up to 100 kGy [12,13]. Radiochromic gel dosimeter based on gelatin and nitro blue tetrazolium chloride (NBT) and evaluated at 527 nm (absorption peak position) indicated stability for up to 2 weeks after irradiation with linear dose range 10–500 Gy [14]. It was stated in [15], that acetonitrile solution containing leuco crystal violet was applicable for dosimetry in the dose range of 50–600 Gy. Another, leuco dye “Black305”-based composite resin dosimeter containing cerium-doped yttrium aluminium perovskite YalO3:Ce (YAP:Ce) scintillating powder mixed with 2-(4- Methoxystyryl)-4,6-bis(trichloromethyl)-1,3,5-triazine (MBTT) was changing color from yellow to black (610 nm) after irradiation up to 120 Gy [16]. It was shown, that mixing of MBTT and Black305 keeping 3.0/3.0 *w/w* ratio, fabricated composite may serve as a fibrous color dosimeter which might be used for measurement of doses up to 80 Gy [17].

Another dosimeter system based on polyacrylamide (PAC) gel enriched with Ag nanoparticles was suggested by Soliman [18,19]. Irradiated gels indicated linear dose response within a dose range of 0–100 Gy. Dose response and color changes due to irradiation were estimated using surface plasmon peak which was observed at 453 nm in UV–Vis spectra.

Widely investigated Fricke dosimeters are usually associated with a good linear dose response in a lower dose range (up to 30 Gy), minimal detectable dose of 0.1 Gy and 0.073 Gy^−1^ cm^−1^ sensitivity [20]. These dosimeters are known for their good reproducibility, but poor post-irradiation stability, especially if kept at higher temperatures: 2 h or longer at 35 °C leads to registration of the higher optical signal and the reconstructed dose differs by up to 7% from the initially delivered dose [21], they are also sensitive to the daily light. If the dosimeters are kept in a light-shielded vacuum case, the initial information could be stored up to 1 month [22]. Dose registration in Fricke dosimeters depends on radiation-induced chemical oxidation of ferrous (Fe^2+^) to ferric ions (Fe^3+^), that can be detected analyzing MRI scans of the solution or using optical techniques [23]. UV–Vis spectroscopy is usually used to analyze Fricke solutions with added ligands, such as xylenol orange which forms chemical complex with ferric ions which is represented by the absorption peak at 585 nm in visible light range [24]. Fricke solution admixed to gelatin water solution, benzoic acid (BA), ferrous ammonium sulfate hexahydrate (FAS), sulfuric acid (SA) and methylthymol-blue sodium salts (MTB) provides a linear dose response up to 10 Gy with good low-dose detectability. The characteristic Fricke gel’s absorption peak at 439 nm is clearly observed in UV–Vis spectra [25]. Magnetic resonance imaging provides R_2_ sensitivity of 0.33 s^−1^ Gy^−1^ for PVA-GTA-MTB compositions in the dose range of 0–10 Gy [26].

More stable as Fricke derivatives are polymer dose gels containing nitro blue tetrazolium chloride as a color indicator. Different authors reported controversial dynamic dose ranges: 2.5–30 Gy [27] and 0–20 mGy for PVA–NBT [28] and for PVA-Ag-NBT [29]. The authors analyzed color change and shift of the absorption peak from 430 to 450 nm for not irradiated and 20 mGy irradiated PVA-NBT gel indicating X-ray reduction induced NBT2+ reduction to mono-formazan (MF+) and formation of stable hydrophobic di-formazan structure in [28] and also additional peak shift to 472 nm [29] due to admixing of silver salt to the same gel and related radiosynthesis of Ag nanoparticles. This interpretation was not approved experimentally, since it was based on processes that are present at higher irradiation doses (several Gy, at least) as compared to those provided by the authors.

Investigation of commercially available PRESAGE^®^ with leuco malachite green (LMG) dye additive indicated dose linearity in the range up to 30 Gy [30]; as well as dynamic range of genipin-based dosimeters (water, gelatin, agarose, genipin, sulfuric acid) was between 0–10 Gy. These gels showed moderate stability up to 600 h [31] and maximum dose sensitivity of 0.035 cm^−1^Gy^−1^ at 10 °C readout temperature [32]. There are some attempts to develop theoretical model for the prediction of color change sensitivity in gels irradiated up to 100 mGy [33].

Performed analysis has shown that the majority of radiation exposure/dose sensors are more or less expensive and sensitive to high doses that are measured in radiation therapy or in an industrial area. There is a lack of a simple individual equipment that would be sensitive enough just to detect low exposure doses to individuals due to occasionally occurring radioactive releases or work in the contaminated environment. Simple and cheap color-changing indicators cannot replace standard dosimeters, but they may help to indicate possible radiation exposure, which might be dangerous for the health of individuals working under irradiation conditions.

## 2. Results

Two different groups of colored thin films produced from PVA-based gels, containing dyes dissolved in organic solvents, or Ag nanoparticles (NPs) produced from AgNO_3_ via radiation-induced synthesis) have been investigated in order to assess their radiation sensitivity in the range of 0–10 Gy with a special focus on gel’s ability to react to low exposure doses.

It is well known that radiation initiates polymerization processes in PVA [34,35], radiation degradation and decolorization of specific dyes [36,37] synthesis of Ag nanoparticles, presence of which is responsible for film color. The main driving force in all these processes is interaction of radiolysis products with corresponding species in gel films. This implies the need for individual and complex approach when analyzing radiation induced changes in gels made of different compositions.

Radiolysis leads to production of varying amounts of H_2_, H_2_O_2_, H^+^, OH^−^, and solvated e^−^ (aq), hydroxyl (OH^●^) and perhydroxyl (HO_2_^●^) radicals and H^●^ depending on the composition of matter and energy transferred. The solvated e^−^ (aq) and H^●^ are the main reductive species, whereas OH^●^ and HO_2_^●^ are the main oxidizing species.

### 2.1. Radiation-Induced Processes in Dyes

The OH^●^ radical is inert with respect to free air oxygen, but is highly reactive towards various functional groups of organic compounds. The hydroxyl radicals (OH^●^) may subsequently react with almost any nearest functional group of organic dyes (RH) according to the below provided scheme and lead to decoloration and final degradation of dyes [38,39,40]: (1)$$\mathrm{RH}+{\mathrm{OH}}^{•}\to R^{•}+H_{2}O$$
(2)$$R^{•}+O_{2}\to {\mathrm{ROO}}^{•}$$
(3)$${\mathrm{ROO}}^{•}+\mathrm{RH}\to \mathrm{ROOH}+R^{•}$$
(4)ROOH→RO+OH
(5)$${\mathrm{RO}}^{•}+\mathrm{RH}\to \mathrm{ROH}+R^{•}$$
(6)$$\mathrm{ROH}+2{\;\mathrm{OH}}^{•}\to {R^{\prime }}^{•}C\left(\mathrm{OH}\right)H)+H_{2}O$$
(7)$${R^{\prime }}^{•}C\left(\mathrm{OH}\right)H)+O_{2}\to {R^{\prime }}^{•}+{\mathrm{CO}}_{2}+{\;H}_{2}O$$

The rate of most of these OH^●^ radical reactions is controlled by diffusion which proceeds more slowly in solid gel films as compared with gel solution. In addition, the electrophilicity of OH^●^ favors attacks on electron-rich units [41] including aromatic units, bridge double bonds, and side groups (Figure 1). It was indicated [42] that the double bond in azo group (N=N) is weak and easy to degrade, as compared with degradation of aromatic rings.

The addition of an OH^●^ to a double bond damages the intramolecular conjugation system, making it shorter. In turn, the unpaired electron delocalizes over the remaining conjugated bonds (including the aromatic unit). As a result, conformational stresses arise in the OH adduct due to the mismatch between the electronic configurations of the OH adduct and the initial unit. Relaxation is possible in a rigid dye molecule [40,42] due to cleavage of C–N, C–O and C–C bonds. These processes lead to the reduction or elimination of color. Depending on the dye composition and relaxation conditions, the OH adduct can avoid bond cleavage, but damage to the intramolecular conjugation system remains. The resulting radical can then decay in the reaction with oxygen or recombine with another radical.

The H-abstraction from the initial hydroxyl groups of the dye is unlikely to make a significant contribution to decoloration, since the content of hydroxyl groups is low, and the resulting radicals are capable of fast regeneration of the hydroxyl group [40,42].

Radiolytic decoloration processes in dyes belonging to the different classes is similar. Decoloration of dyes proceeds when OH^●^ radicals interact with double bonds that are responsible for the conjugation of atoms into a combined chromophore system and for the color of the dye. Further development of resulting radicals and their derivatives are responsible for degradation of dye [42].

It should be noted that especially for azo dyes, the products formed in the reactions of the OH^●^ and dye molecules were found to be highly reactive towards OH^●^ radicals. For these reasons, the efficiency of OH^●^ radicals in decolorization was relatively low. In contrast to this, solvated electron $e_{\mathrm{aq}}^{-}$ plays an important role in the destruction of the intense color (color giving centre) of azo dye. This is due to the fact that the reaction of a solvated electron with azo group (N=N) in unreacted molecules is very fast and produced radical anion which quickly protonates, forming hydrazyl radical; however, the reactivity of $e_{aq}^{-}$ with the transformed molecules is rather low [36]:(8)$$-N=N-+e_{\mathrm{aq}}^{-}\to -N-N^{-}\to \left(+H_{2}O\right)\to -N^{●}-\mathrm{NH}-+{\mathrm{OH}}^{-}$$

The same radical forms in H^●^ addition reaction:(9)$$-N=N-+H^{●}\to -N^{●}-\mathrm{NH}-$$

Both solvated electron and H^●^ addition leads to the destruction of -N=N- bond and consequently to decolorization [41].

### 2.2. Radiation Induced Processes in PVA

In the absence of radical scavenger, OH^●^ radicals can induce the radiation crosslinking of PVA molecules [2,34,41,43].
(10)$$\mathrm{PVA}\left(H\right)+{\mathrm{OH}}^{•}\to {\mathrm{PVA}}^{•}+H_{2}O$$
(11)$$2{\mathrm{PVA}}^{•}\to \mathrm{PVA}-\mathrm{PVA}$$

The hydroxyl radicals almost exclusively react with PVA; however, other polymerization routes are also possible:(12)$$2\mathrm{PVA}+2H^{•}\to {\mathrm{PVA}}^{•}+2H_{2}$$

### 2.3. Radiation-Induced Synthesis of Ag NPs

The solvated $e_{\mathrm{aq}}^{-}$ and H^●^ are responsible for the reduction of silver ions Ag^+^ from metal salt (AgNO_3_) solution to neutral silver atoms (Ag^0^):(13)$${\mathrm{Ag}}^{+}+e_{\mathrm{aq}}^{-}\to {\mathrm{Ag}}^{0}$$
(14)$${\mathrm{Ag}}^{+}+H^{*}\to {\mathrm{Ag}}^{0}+H^{+}$$

Due to the high electron capture cross-section for Ag^+^, a large number of Ag^0^ atoms can be produced in irradiated silver salts (Ag NO_3_ in our case). The Ag^0^ atoms may join with Ag^+^ ions, thus formatting silver nanoparticles:(15)$${\mathrm{Ag}}_{n}+{\mathrm{Ag}}^{+}\to {\mathrm{Ag}}_{n+1}^{+}$$
(16)$${\mathrm{Ag}}_{n+1}^{+}+e_{\mathrm{aq}}^{-}\to {\mathrm{Ag}}_{n+1}$$
(17)$${\mathrm{Ag}}_{n+1}^{+}+H^{*}\to {\mathrm{Ag}}_{n+1}+H^{+}$$

The polyvinyl alcohol (PVA) chain plays a significant role in avoiding the formation of metal hydroxide clusters by hydrolysis of metal ions, thus preventing them from aggregation, since hydroxyl (–OH) groups in the structure of the PVA are capable of absorbing metal ions (Ag^+^) through secondary bonds according to the reaction:(18)$$\mathrm{PVA}-\mathrm{OH}+{\mathrm{Ag}}^{+}\to \mathrm{PVA}-O-{\mathrm{Ag}}^{0}+H^{+}$$

PVA* radicals and solvated $e_{aq}^{-}$ being the most important PVA radiolysis products are also participating in the reduction reactions of silver ions (Ag^+^):(19)$${\mathrm{Ag}}_{2}^{+}+e_{\mathrm{aq}}^{-}/{\;\mathrm{PVA}}^{.}\to {\left(\mathrm{Ag}\right)}_{n}$$

It should be noted that radiation-induced reactions in gel solutions proceed very quickly: the radicals are produced in a very short time (10^−12^−10^−6^ s) and the radicals may react within ≥10^−6^ s. Due to a significantly reduced amount of water in gel films, the reactions are much slower since free radicals are relatively stable and radiation-induced processes proceed via slow (seconds) radical migration.

Two different groups of colored thin films produced from PVA-based gels, containing dyes dissolved in organic solvents, or Ag NPs produced from AgNO_3_ via radiation induced synthesis have been investigated in order to assess their radiation sensitivity in the range of 0–10 Gy, with a special focus on gel’s ability to react to low exposure doses.

Analysis of radiation-induced changes of optical characteristics of 6 MeV irradiated PVA gel films was performed using the results of UV–Vis spectroscopy.

### 2.4. Irradiated PVA-Dye Gel Films

Depending on the chromophores, the investigated dyes can be classified in two different classes: **Azo dyes** (brilliant carmoisine, brilliant scarlet (red ponceau), methyl red, methylene blue (heterocyclic aromatic azo dye with benzene rings), xylenol orange (sulfonated azo dye) and **Triphenylmethane (triarylmethane) dyes** (brilliant green and brilliant blue).

UV–Vis absorbance spectra of the irradiated PVA gel films prepared with various dyes and the corresponding radiation sensitivity curves, chemical structure and irradiated film sample with corresponding dose are provided in Figure 2, Figure 3, Figure 4, Figure 5, Figure 6, Figure 7 and Figure 8 (see also Table 1 for chemical composition of samples).

Decolorization efficiency/color bleaching effect in all 6 MeV irradiated PVA-dye gel films is shown in Figure 9.

The lowest bleaching effect (5%) was found for PVA-BG film. The bleaching was almost independent from the radiation dose. The highest bleaching effect was obtained from PVA-MR films.

### 2.5. Irradiated Silver-Enriched PVA Films

UV–Vis absorbance spectra of the irradiated silver-enriched PVA gel films, and irradiated film with corresponding dose are provided in Figure 10, Figure 11, Figure 12 and Figure 13 (see also Table 2 for chemical composition of samples).

## 3. Discussions

### 3.1. PVA-Dye Films

There Azo dyes are chemical compounds bearing the functional group R–N=N–R’ in which R and R’ are aryl groups. Because of the electron delocalization through the -N=N- group these compounds have vivid colors. The color is dependent on the chromophore (one or several azo groups connecting aromatic units) and the extent of conjugation. Depending on the number of azo groups there are mono-, di- and triazo-dyes. The following dyes were admixed to PVA to form gel films: brilliant carmosine (mono-azo), brilliant scarlet (ponceau 4R), methyl red (mono-azo); methylene blue (heterocyclic aromatic azo dye with benzene rings belonging to thiazin dye subgroup); xylenol orange (sulfonated azo dye).

The function of the chromophore group in triphenylmethane (triarylmethane) dyes is performed by a quinoid unit formed by introducing an amino or hydroxy group into the para position to the central methane carbon. PVA gel films with admixed brilliant green and brilliant blue dyes have been investigated.

### 3.2. UV-VIS Absorption in PVA-Dye Films

Performed analysis of UV-VIS absorbance spectra has shown that the characteristic absorbance peaks of all PVA dye films were slightly shifted to the longer wavelengths, as compared to the original dyes, see Table 1. The same tendency was observed not only for azo dyes, but also for tryarilmethane dyes. Taking into account that the investigated gel films were polymer composites containing PVA admixed dyes, and that the original absorbance peaks were measured for dye solution only, the shift of absorbance peaks in the films can be attributed to radiation-induced polymerization of PVA which is characterized by absorbance maximum at ~500 nm. The shifts of absorbance peaks of PVA-dye films caused by irradiation were not observed or were negligibly small.

Performed investigation has shown that the sensitivity of irradiated films was higher for the low doses (0.1–2.0 Gy). Estimated dose sensitivity was 0.057 Gy^−1^ for PVA-MB, 0.09 Gy^−1^ for PVA-MR, 0.1 Gy^−1^ for PVA-BC, 0.1 Gy^−1^ for PVA-XiO, 0.15 Gy^−1^ for PVA-BS, as well as 0.065 Gy^−1^ for PVA-BG and 0.4 Gy^−1^ for PVA-BB films.

In the higher dose range, the dose sensitivity of methylene blue (MB) and of methyl red (MR) films remained the same–0.057 Gy^−1^ and 0.09 Gy^−1^, respectively. However, for other PVA-dye films, the dose sensitivity in the dose range > 1.0 Gy appeared as significantly reduced and was 0.004 Gy^−1^ for PVA-BG, (negligible), 0.004 Gy^−1^ for PVA-XiO (negligible), or 0.0125 Gy^−1^ for PVA-BS and 0.025 Gy^−1^ for PVA-BC. The highest sensitivity of 0.033 Gy^−1^ in this dose range was found again for PVA-BB films, indicating the potential of these films for applications in low-dose radiation dosimetry.

### 3.3. Sensitivity of PVA-Dye Films

In general, decreasing absorbance of irradiated dyes in the visible range depends on the attacks of primary intermediates (OH^●^, H^●^ and $e_{\mathrm{aq}}^{-}$) to the chromophoric part of the azo-dye molecule, (e.g., –C–N=N–C– bridge connecting two aromatic groups) resulting in an irreversible reaction, which leads to the destruction of the extensive conjugated electronic system of chromophore and formation of hydrazyl-type radicals by adding the radical to N=N group. The combination of these radicals causes saturation of the N=N and decolorization of the solution. Observed relatively high PVA-mono azo dye films sensitivity in the initial stage of radiation (irradiation time < 2 min, irradiation doses < 2 Gy) might be caused by the contribution of all three intermediates to the destruction of N=N bonds. When the irradiation starts, it generates a number of radiolysis products, including OH^●^, H^●^ and $e_{\mathrm{aq}}^{-}$ in PVA-dye gel films. Radiolysis in polymeric films proceeds more slowly than in water solutions due to significantly reduced water content in the film; however, it was found to be efficient enough to generate reasonable amount of H^●^ and $e_{\mathrm{aq}}^{-}$ in few minutes [44]. Since the reactions of solvated electrons and H^●^ atoms with azo groups are very fast, significant number of azo groups that are mainly responsible for the dye’s color can be destroyed in a very short time, thus indicating high decoloration efficiency. Decolorizaion efficiency of the solvated electron and H^●^ atom is almost 100% in the initial stage (up to few minutes) of irradiation [41]. At longer irradiation times (higher doses), however, the dye is already depleted and the higher proportion of solvated electrons $e_{\mathrm{aq}}^{-}$ and H^●^ atoms in irradiated gel film may not react with the azo moiety causing decolorization, but take part in other reactions. The role of hydroxyl OH^●^ radical in the decolorization of PVA-azo dye films was modest in the whole investigated dose range, since the products formed in OH^●^ reactions with dye molecules were highly reactive towards OH^●^ radicals. It should be also noted that the decolorization process was affected by the competing PVA polymerization process in which hydroxyl radical plays a significant role (see Equations (4) and (5)).

It should be noted that the PVA-dye film dose sensitivity depends also on the molecular structure of the dye. This could be demonstrated, analyzing the reasons why the PVA-BB films indicated the highest and PVA-MB films the lowest dose sensitivity among all investigated polymer-dye gel films.

Brilliant blue is highly aromatic triphenylmethane dye. Usually, degradation of gels belonging to this group proceeds via two competing processes: N-demethylation and the destruction of the conjugated structure. BB is highly aromatic compound having many attacking sides. Therefore, the addition of OH^●^ radicals is more favorable than the N-demethylation process during the initial stages of the degradation and the polyaromatic rings are completely destroyed at the initial stage of radiation. The decrease in UV–Vis absorbance might be caused by oxidative degradation of BB and generation of reactive oxidants ${\mathrm{SO}}_{4}^{-}$ and OH^●^. Further reactions with these oxidants lead to the cleavage of benzene rings into the smaller organic molecules [45,46].

Methylene blue (MB) dye is a heterocyclic aromatic azo dye with benzene rings belonging to the thiazin dye subgroup. Due to its molecular structure, MB is more stable, as mono azo dyes are, and the destruction of MB is more complicated. The dye’s color depends on its chromophore (N-S conjugated system on the central aromatic heterocycle) and auxochrome (N-containing groups with lone pair of electrons on the benzene ring) groups. Degradation of MB proceeds via demethylation, braking of the central and the side aromatic rings and conversion of the fragments produced from the first two steps to intermediate species. MB loses its color due to degradation of aromatic rings. The main contribution to decolorization (ring opening) is done by OH^●^ which attacks the C-S^+^=C functional group leading to its transformation in C-S(=O)-C. In order to conserve double bond conjugation, which is lost through this transformation, the central aromatic ring containing heteroatoms S and N is opened. Hole-induced H^+^ participates in formation of CH and NH bonds. Such splitting of a complex molecule into a smaller and highly oxidized intermediate is the primary reason for dye degradation [47].

### 3.4. UV–Vis Absorption in Silver-Enriched PVA Films

It is known that the presence of metal NPs may increase photon absorption ability of the polymer composite. Formation of silver NPs in PVA-AgNO_3_ gel proceeds due to spontaneous protonation of PVA during the drying process or due to metal nanoparticle generation inside PVA upon irradiation. As a result of radiation exposure, one proton is released for every Ag^+^ ion that is reduced to a neutral Ag^0^ atom [48]. This contributes to the color changes of AgPVA films from light yellow to yellow brown. Summarizing information on radiation-induced chemical transformations and silver reduction in irradiated Ag-PVA samples was provided in our previous article [2]; however, more details on radiation-induced Ag-PVA gel composites formation could be found in [49,50].

Analysis of UV–Vis absorbance spectra of 6 MeV irradiated silver-enriched PVA films indicated presence of the absorbance peaks that were visible at 424 nm for 0.21 wt% AgNO_3_ concentration PVA films and at 436 nm for samples containing 1.01 wt% of silver nitrate. These peaks were attributed to the local surface plasmon resonance (LSPR) phenomena, which may be detected due to the presence of metal NP in polymer composite film. Positions at which both absorption peaks were observed were very close to the wavelength of 430 nm, which is characteristic for spherical silver NPs. The initial peak at zero dose was related to the precipitation of Ag nano seeds during the gel’s drying process. It was found that the LSPR peak intensity of the irradiated samples was increasing with the irradiation dose, indicating the increasing number of the produced silver NPs. LSPR peak intensity was dependent on Ag concentration and increased concentration led to higher radiation sensitivity 0.11 Gy^−1^ comparing to 0.068 Gy^−1^. Since well-defined LSPR peaks were observed also after irradiation of films, it was suggested that there was no agglomeration of Ag particles. A very small, almost negligible shift of the LSPR peak towards higher wavelengths was indicated, which supported the initial suggestion that the size of AgNPs remained the same after irradiation.

In order to improve polymers solubility in water, some other solvents were introduced to the hydrogels prepared using 0.21 wt% AgNO_3_ concentration. Analysis of UV–Vis spectra has shown a plasmonic peak at 437 nm for films, containing ethanol and LSPR peak at 440 nm for films containing isopropyl alcohol. Both identified peaks were close to the characteristic position (430 nm) of LSPR for spherical Ag NPs. The LSPR peak intensity was higher for the films, containing isopropanol; however, radiation sensitivity of both 1AgETAPVA and 1AgISOPVA films was almost the same: 0.1197 Gy^−1^ and 0.1198 Gy^−1^, respectively.

## 4. Conclusions

It was found that PVA-dye gel films in general may serve as possible radiation sensors for occasional or accidental dose registration in the range up to 10 Gy.

Radiation sensitivity of films as evaluated from UV–Vis spectra was dependent on specific parameters (dyes concentration, molecular structure, energy applied) and varied between 0.057 Gy^−1^ and 0.40 Gy^−1^. PVA-dye films indicated higher sensitivity in the initial/low-dose (up to 1 or 2 Gy) irradiation stage.

The highest sensitivity of 0.4 Gy^−1^ was found for the irradiated PVA-BB (Brilliant blue) films and the lowest–0.057 Gy^−1^ for the PVA-MR (Methyl red) in the dose interval between 0 and 2 Gy^−1^. The sensitivity in the higher dose interval was rather modest.

PVA-MR films indicated permanent tendency for radiation induced bleaching. The final decolorization was up to 33.3% after irradiation of films 10 Gy dose.

In general, polymer composites enriched with Ag nanoparticles were more sensitive than the PVA-dye films. The radiation sensitivity of these films was dependent on silver salt concentration in the initial gel composition. Exchange of some water with ethanol or isopropanol in the initial polymer gel’s composition, aiming at better solubility of PVA, contributed also to the enhancement of the film’s radiation sensitivity which was found to be in the order of 0.12 Gy^−1^. It should be noted that this sensitivity was achieved for the films that were produced using low AgNO_3_ concentration gel.

A color increase by 30–40% was found for Ag-PVA films irradiated with 6 MeV photons to 10 Gy dose.

Detailed investigation of low-dose (<1–2 Gy) impact on film color belongs to the future actions towards development of the simple flexible colored polymer films dose indicators that could be attached to the garment surface in order to protect individuals from occasional occupational exposure.

## 5. Materials and Methods

### 5.1. Materials

Polyvinyl alcohol (PVA) hydrogels are widely used as a hosting material for additives used in the development of radiation sensors. PVA films are almost transparent in the UV–Vis region, chemically resistant and thermostable. Additionally, this biodegradable, biocompatible polymer is known for low cytotoxicity.

For the preparation of PVA-dye gel films and PVA-Ag films, corresponding materials were purchased from different suppliers: (Poly(vinyl) alcohol, (C_2_H_4_O)_n_, M_w_ ~31,000, Sigma-Aldrich Chemie GmbH, Regensburg, Germany), silver nitrate (AgNO_3_, purity ≥ 99.0%, Sigma-Aldrich Chemie GmbH, Regensburg, Germany), ethanol (C_2_H_5_OH, purity ~96%, Euro-Chemicals GmbH, Nordhorn, Germany), and isopropanol ((CH_3_)_2_CHOH, with purity ≥ 99.0%, Euro-Chemicals GmbH, Nordhorn, Germany).

Chemical compositions of fabricated PVA-dye gel films with indicated dye suppliers are provided in the Table 1, as well as chemical composition of PVA-Ag films, in Table 2.

### 5.2. Preparation of PVA-Dyes Films

Dye powder was dissolved in ethanol and 5% of BB, BC, BG, BS or 10% of MB, MR, XiO dye solutions have been prepared. PVA powder in different proportions was dissolved in the distilled water under continuous stirring in magnetic stirrer keeping 60 °C temperature. 8% and 10% PVA–water solutions have been prepared and left to cool down. Then the prepared 400 µL of dye–ethanol solutions were mixed with corresponding PVA–water solutions (as is indicated in Table 1) and stirred continuously at room temperature. Prepared PVA-dye gels were casted in Petri dishes to form series of thin layer(film) samples containing different dyes and left to dry at room temperature in dark for 72 h. Dried samples containing a reduced amount of water were slightly heated and removed from Petri dishes making films ready for the investigation.

Measured thickness of the prepared films was ~0.1 mm.

**Table 1 gels-09-00189-t001:** Chemical composition of prepared different PVA-dye films.

Dosimeter	Dye	Molecular Formula	Composition
1	Brilliant blue FCF, BB (analytical standard, Merck KGaA, Darmstadt, Germany)	C_37_H_34_N_2_Na_2_O_9_S_3_	400 µL 5% (BB in ethanol solution) + 20 g 8% PVA solution
2	Brilliant carmoisine (E122), BC (Merck KGaA, Darmstadt, Germany)	C_20_H_12_N_2_Na_2_O_7_S_2_	400 µL 5% (BC in ethanol solution) + 20 g 8% PVA solution
3	Brilliant green, BG (Merck KGaA, Darmstadt, Germany)	C_27_H_34_N_2_O_4_S	400 µL 5% (BG in ethanol solution) + 20 g 8% PVA solution
4	Brilliant scarlet/Ponceau 4R), BS (Merck KGaA, Darmstadt, Germany)methylen	C_20_H_11_N_2_Na_3_O_10_S_3_	400 µL 5% (BS in ethanol solution) + 20 g 8% PVA solution
5	Methylene blue, MB (Merck KGaA, Darmstadt, Germany)	C_16_H_18_ClN_3_S	400 µL 10% (MB in ethanol solution) + 20 g 8% PVA solution
6	Methyl red, MR (Merck KGaA, Darmstadt, Germany)	C_15_H_15_N_3_O_2_	400 µL 10% (MR in ethanol solution) + 20 g 10% PVA solution
7	Xylenol orange disodium salt, XiO ((Merck KGaA, Darmstadt, Germany)	C_31_H_32_N_2_Na_2_O_13_S	400 µL 10% (XiO in ethanol solution) + 20 g 10% PVA solution

### 5.3. Preparation of PVA Films with Ag Nanoparticles

Similar preparation method was used to make thin PVA films enriched with Ag particles. A certain amount (see Table 2) of AgNO_3_ was added drop by drop to the PVA–water solution during the stirring and two sets of PVA-AgNO_3_ gels with two different AgNO_3_ concentration has been prepared. Some part of water in gel’s solution was replaced by ethanol in order to increase the solubility of PVA in order to assess the effectiveness of ethanol and isopropanol in preventing of synthesized metal nanoparticles from agglomeration; a certain amount of these solvents was added drop by drop to the ready PVA–AgNO_3_ mixture. Two additional sets of PVA-AgNO_3_ gels were prepared. Prepared solutions were casted in Petri dishes for the formation of PVA-Ag films and left to dry at room temperature in dark for 72 h. Dried films were removed from the Petri dishes and made ready for the investigation. It should be noted that initial colorless samples became light yellow through the drying process due to possible formation of silver seeds [2].

**Table 2 gels-09-00189-t002:** Chemical composition of prepared PVA films with Ag.

Material	Chemical Composition of Polymer Films, wt%				
	PVA	1AgPVA	AgPVA	1AgIsoPVA	1AgEtaPVA
AgNO_3_		0.21	1.01	0.21	0.21
PVA (C_2_H_4_O)_n_	20	19.76	18.92	19.41	19.35
Ethanol CH_3_CH_2_OH					2.10
Isopropyl alcohol (CH_3_)_2_CHOH				2.53	
H_2_O	80	80.03	80.07	77.85	78.34

### 5.4. Irradiation and Optical Analysis Techniques

Prepared PVA-dye and PVA-Ag films were irradiated with 6 MeV X-ray photons in a medical linear accelerator, Clinac DMX (Varian Medical Systems, Inc., Palo Alto, CA, USA). In order to deliver proper doses, experimental films were placed between two boluses (9 cm PMAA bolus at the bottom and 1 cm on the top) keeping 100 cm SSD. Irradiation doses from the range 0.1–10.0 Gy were applied during this investigation.

To analyze the color change of the developed dose indicators, ultraviolet–visible light (UV–Vis) spectroscopy was chosen. UV–Vis spectra of irradiated films were obtained using spectrophotometer Ocean Optics with USB4000 (Ocean Optics, Inc., Dunedin, FL, USA). Optical characteristics were analyzed in the range of 300–900 nm using “Ocean View” spectroscopy software, product version 1.6.7 by Ocean Optics.

Two parameters are important in analyzing the applicability of PVA-based colored films for dosimetry issues: dose sensitivity of films and color-changing factor in irradiated films.

Dose sensitivity (Gy^−1^) is usually expressed as the slope of the curve indicating the film’s UV–Vis absorption intensity peak response to the irradiation dose.

Color changes in the irradiated samples were evaluated using following formulas: [44]
(20)$$\mathrm{Color}\;\mathrm{increase}=\frac{A-A_{0}}{A_{0}}\times 100\%,$$
(21)$$\mathrm{Color}\;\mathrm{bleaching}=\frac{A_{0}-A}{A_{0}}\times 100\%,$$
where A_0_ is the film absorbance at zero dose and A is film absorbance after irradiation with a certain dose.

## Figures and Tables

**Figure 1 gels-09-00189-f001:**
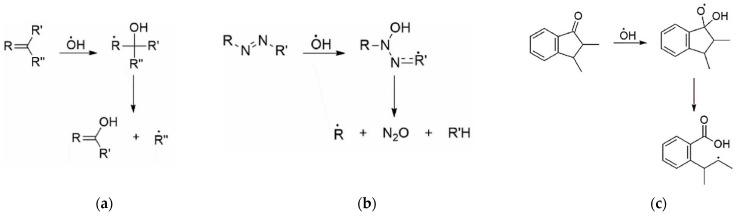
Interaction of OH^●^ with: aromatic units (**a**), azo bond (**b**), chromophores (**c**).

**Figure 2 gels-09-00189-f002:**
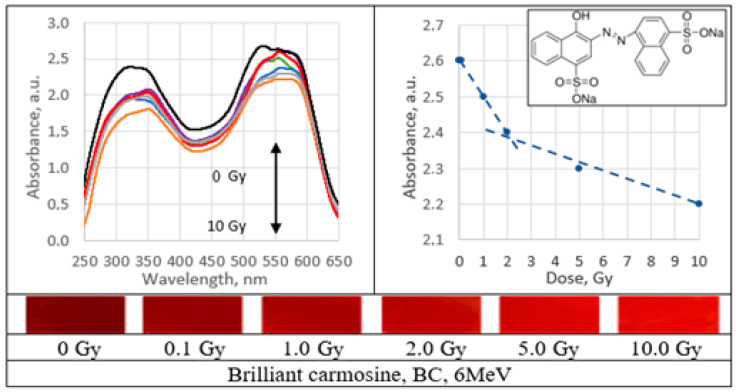
Absorbance spectra and sensitivity of the irradiated brilliant carmoisine films.

**Figure 3 gels-09-00189-f003:**
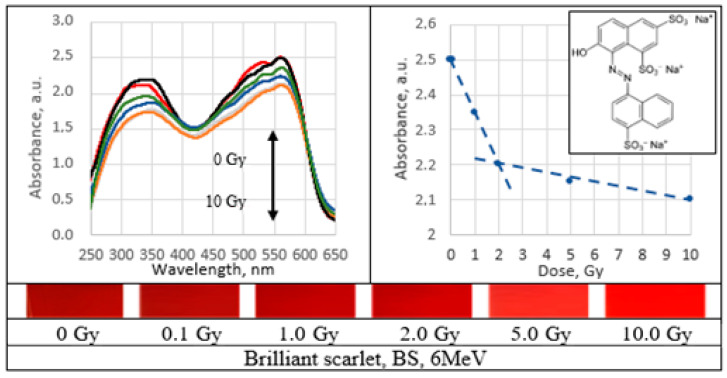
Absorbance spectra and sensitivity of the irradiated brilliant scarlet films.

**Figure 4 gels-09-00189-f004:**
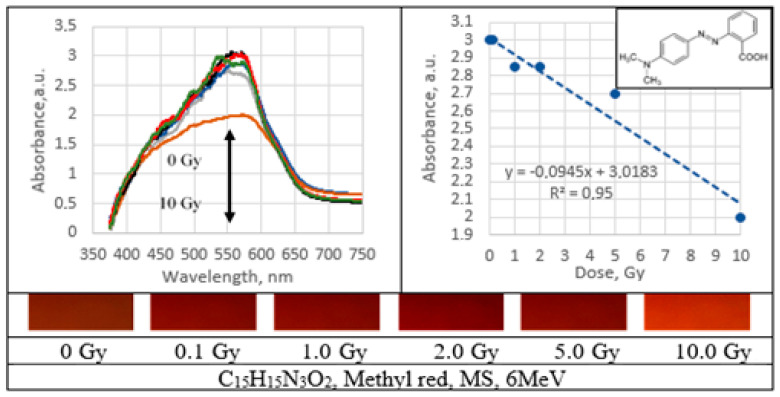
Absorbance spectra and sensitivity of the irradiated methyl red films.

**Figure 5 gels-09-00189-f005:**
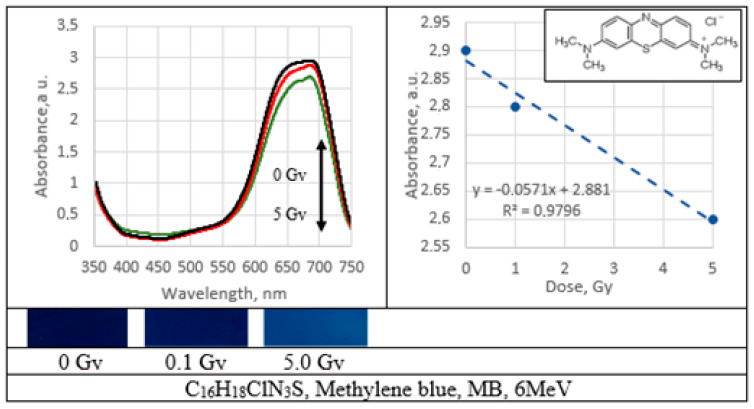
Absorbance spectra and sensitivity of the irradiated methylene blue films.

**Figure 6 gels-09-00189-f006:**
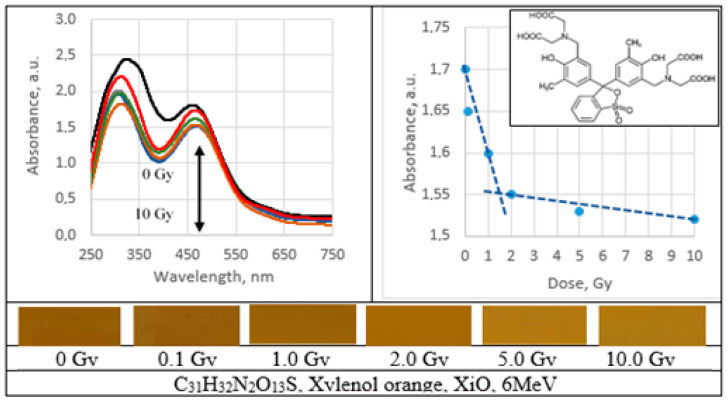
Absorbance spectra and sensitivity of the irradiated xylenol orange films.

**Figure 7 gels-09-00189-f007:**
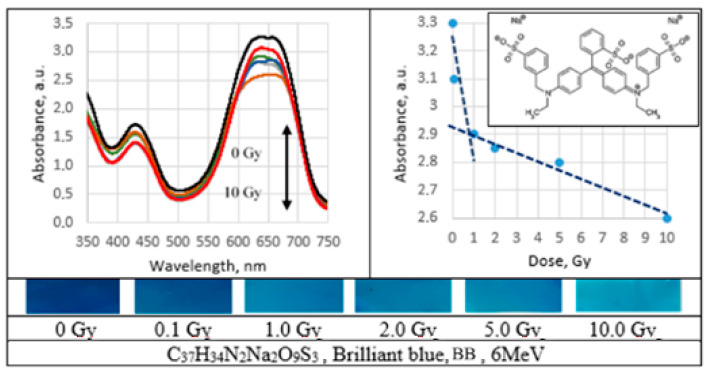
Absorbance spectra and sensitivity of the irradiated brilliant blue films.

**Figure 8 gels-09-00189-f008:**
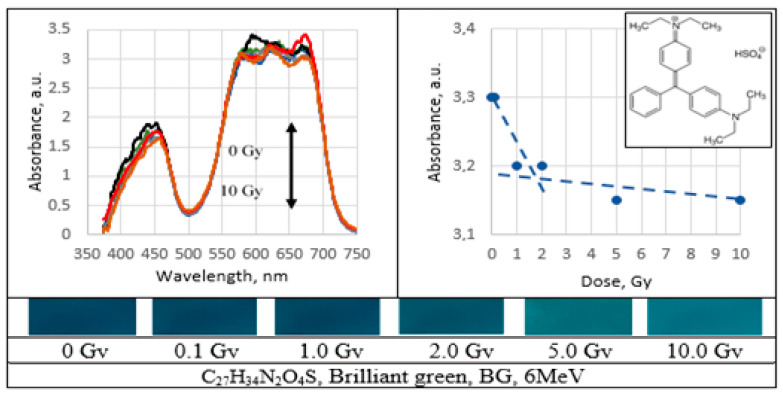
Absorbance spectra and sensitivity of the irradiated brilliant green films.

**Figure 9 gels-09-00189-f009:**
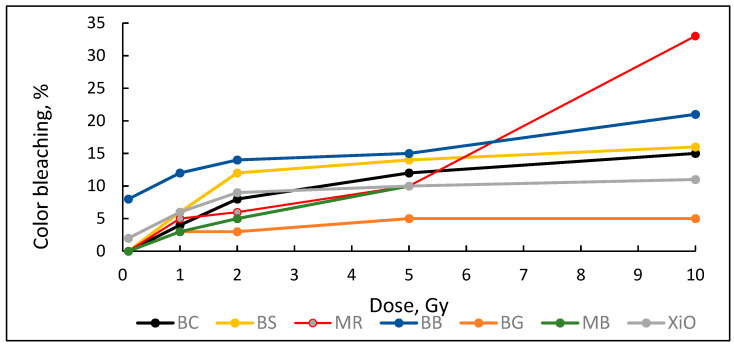
Color bleaching effect in irradiated investigated PVA-dye films. (Dose axis starts from 0.1 Gy.)

**Figure 10 gels-09-00189-f010:**
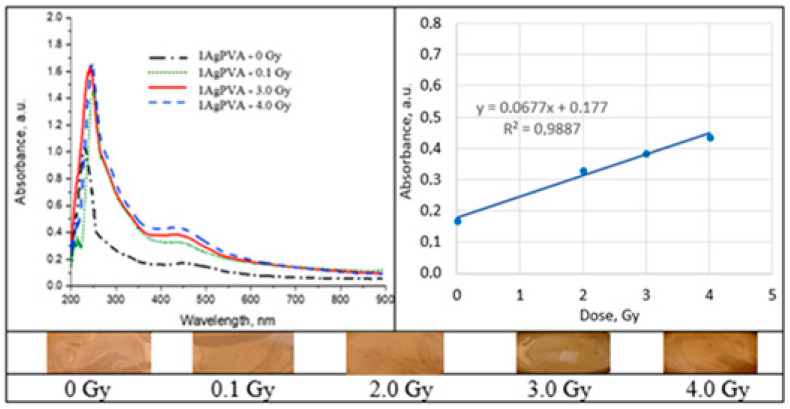
Absorbance and sensitivity of the 6 MeV irradiated 1AgPVA films.

**Figure 11 gels-09-00189-f011:**
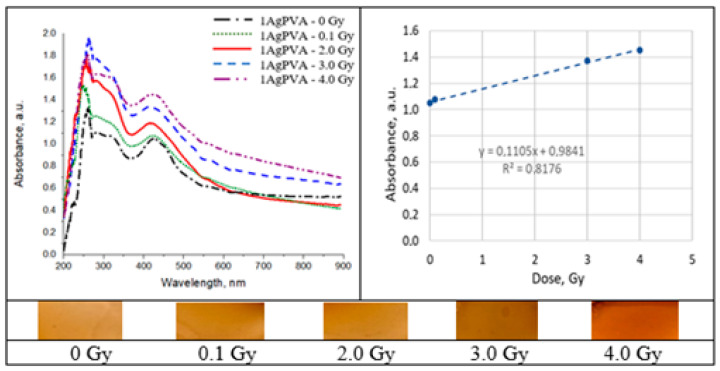
Absorbance and sensitivity of the 6 MeV irradiated AgPVA films.

**Figure 12 gels-09-00189-f012:**
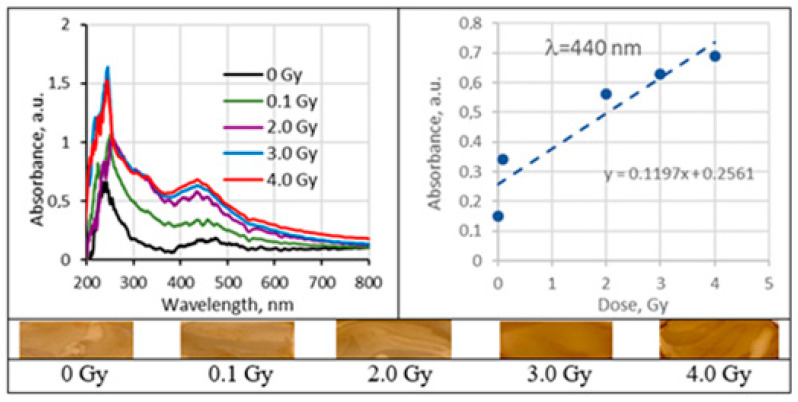
Absorbance and sensitivity of 6 MeV irradiated 1AgEthaPVA films.

**Figure 13 gels-09-00189-f013:**
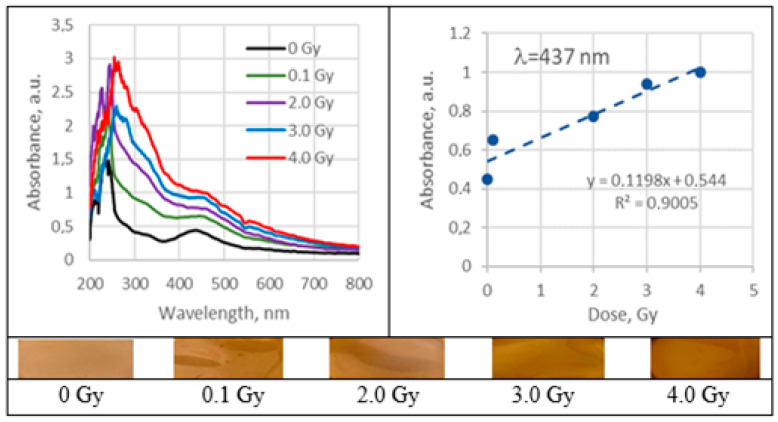
Absorbance and sensitivity of 6 MeV irradiated 1AgEthaPVA films.3. Discussions.

## Data Availability

Data is available by corresponding author upon request.

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
