# Peer review of "Investigation of Colored Film Indicators for the Assessment of the Occasional Radiation Exposure"

_gels, 2023, doi:10.3390/gels9030189_

Round 1

Reviewer 1 Report

Dear Authors

Thanks for your investigation and apply to Journal

Manuscript is well generally but need some correction and revision

More recent studies can be referenced in the study.

The page layout should definitely be reviewed and it would be more appropriate to reschedule.

Visual materials and graphical summary can be suggested.

Flow should be controlled in English. Minor typos should be corrected.

The first use of abbreviations should be descriptive. Abbreviations can be used in text after long naming

this evalution details are as below:

This work; He studies the effects of occupational radiation exposure.
It discusses methods for examining, measuring and reducing the effects of radiation exposure. The results obtained with the development of the developed polymer-based material and the experiments are discussed.
The concept of the article and the methods it applies are appropriate and sufficient.
Some changes can be made to reader interest and content.

The summary is too long. The summary should focus on providing more information with a shorter narrative. Reference notation is a situation that should be avoided in the abstract. The summary should be study-specific, descriptive, problem posing, and focused on results.
Some of the information presented in the summary can be explained in the introduction.

In the introduction, some quotations are kept uxun. It may be appropriate to select references from more recent studies (last 3 years) and journals with high IF.

There is no Method title in the study. The method needs to be rewritten in a clear, understandable way and according to scientific principles. In the method section, it should be emphasized why the method was chosen and the relevance of the chosen method to the problem. Information such as standards, scientific laws, technological equipment used is missing, and these should be explained with pictures.

The results part is descriptive, but this part should be divided into subsections. The theoretical part can be explained under one heading, and the experiment and its results can be explained under another subheading. Chart descriptions should be presented above or below the chart.

The discussion part is too long.

The article organization can be rearranged in the study. Knowledge-intensive work makes reading difficult. Although the number of pages is long, the number of resources used is insufficient.

The work should include more chapters and lat chapters. Not all information should be presented at once. 

Reviewer 2 Report

In this manuscript, the authors prepared a series of hybird film materials consisting of PVA gel and organic dyes. These hybrid films show excellent solour indicative property for the assessment of the occasional radiation exposure. The work is interesting and the writing is in well scientific English. The referee supports its publication after considering the following issues.

1 The abstract should be more concise. Currently it is too long.

2 The interaction between PVA and organic dyes should be investigated by FTIR.

3 Is the colour change visible by eyes? It is better to show some figures or videos as supporting information.

4 The authors should further explain why different dyes gives different sensitivity.

5 With radiation exposure, is the organic dyes degraded?

6 Could the films be recyced and reused?
